# IN DEFENSE OF DUAL-ENCODERS FOR NEURAL RANKING

## ABSTRACT

Transformer-based models such as BERT have proven successful in information retrieval problem, which seek to identify relevant documents for a given query. There are two broad flavours of such models: *cross-attention* (CA) models, which learn a joint embedding for the query and document, and *dual-encoder* (DE) models, which learn separate embeddings for the query and document. Empirically, CA models are often more accurate, which has motivated a series of works seeking to bridge this gap. However, a fundamental question remains less explored: does this performance gap reflect a limitation in the *capacity* of DE models, or in the *training* of such models? In this paper, we study this question, with three contributions. First, we establish theoretically that with a sufficiently large encoder size, DE models can capture a broad class of scores without cross-attention. Second, we show empirically that on real-world problems, the gap between CA and DE models may be due to the latter *overfitting* to the training set. To mitigate this behaviour, we propose a distillation strategy that focuses on preserving the *ordering* amongst documents, and confirm its efficacy on benchmark neural re-ranking datasets.

## 1 INTRODUCTION: TRANSFORMER-BASED NEURAL RANKING

Information retrieval (Mitra & Craswell, 2018) is the classic problem of identifying relevant documents for a given query. Typically, such retrieval is performed in a two-step manner (Matveeva et al., 2006): one uses an initial model to efficiently *retrieve* a candidate set of documents for a query, and then uses a second model to *re-rank* these candidates. The retrieval stage is generally implemented by a model with low inference cost, for which candidate document generation is tractable; if this candidate set is reasonably small, it is feasible to use more complex models in the re-ranking stage.

Transformer-based models such as BERT (Devlin et al., 2019) have proven successful in both the retrieval and re-ranking stages. Such *neural ranking* models have two flavours: *dual-encoder* (*DE*) models (Lee et al., 2019; Chang et al., 2020; Karpukhin et al., 2020; Xiong et al., 2021; Luan et al., 2021), which learn separate (factorised) embeddings for the query and document; and *cross-attention* (*CA*) models (Nogueira & Cho, 2019; Dai & Callan, 2019; Yilmaz et al., 2019; MacAvaney et al., 2019; Gao et al., 2020), which learn a joint embedding for the query and document. Only DE models are applicable for retrieval, as they admit efficient nearest neighbour search (Guo et al., 2020; Johnson et al., 2021) to identify candidate documents. Both CA and DE models are applicable for re-ranking; however, empirically, CA models have proven more accurate (Hofstätter et al., 2020a).

Does the re-ranking performance gap between CA and DE models reflect a limitation in the inherent *capacity* of the latter's factorised representation, or in the *training* of DE models? These questions are of conceptual interest, but are also practically relevant: DE models allow for more efficient inference owing to the ability to pre-compute document embeddings, which enables fast similarity search when presented with a candidate query (Khattab & Zaharia, 2020; Hofstätter et al., 2020b). Several works have explored means of improving DE models, including by changing[1] the scoring layer (Khattab & Zaharia, 2020; MacAvaney et al., 2020; Hofstätter et al., 2020b), and by *distilling* predictions from a CA model (Lu et al., 2020; Izacard & Grave, 2020; Yang & Seo, 2020; Hofstätter et al., 2020a; Miech et al., 2021). However, the root cause of the gap between CA and DE models remains elusive.

In this paper, we study these questions, with the following contributions:

---

[1]Unless otherwise stated, we use "DE" to mean a dual-encoder with dot-product scoring per Equation 2.

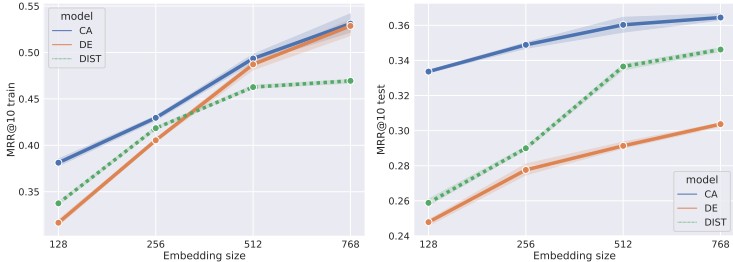

Figure 1: Comparison of cross-attention (CA) and dot-product based dual-encoder (DE) models on the MSMARCO-Passage re-ranking task. Using 6-layer BERT models with varying embedding size (Turc et al., 2019), we report the train and dev set MRR@10 averaged over three independent trials. For sufficiently large embedding dimension, the DE model closely matches the performance of the CA on the *training* set; however, there is a sizable gap in the *test* set performance. This points to the poorer DE model performance being largely an issue of generalisation, rather than model capacity. Suitable distillation (DIST) from the CA model manages to prevent such overfitting — potentially by *worsening* the training performance — and largely bridges the gap between the two models. See §3.2 for details on the experimental setup, and §4 for details on our distillation strategy.

(i) we establish theoretically that with a sufficiently large embedding dimension (and under mild assumptions), any continuous ground-truth scores can be modelled by generic DE models (Proposition 1), and in particular by sufficiently deep transformer-based DE models (Proposition 2). Thus, in principle, there is no fundamental restriction in using DE versus CA models for re-ranking.

(ii) we show empirically (Figure 1) that on real-world problems, CA and DE models can achieve similar *training* performance, but DE models may do worse on *test* data due to *overfitting* (§3.2). Thus, the DE models may suffer due to poorer *generalisation* ability rather than *capacity*.

(iii) to mitigate the above, we propose a distillation strategy focussing on mimicking the teacher's *ordering* amongst documents (§4). This generalises the recent margin MSE loss (Hofstätter et al., 2020a) (§4.1), and justifies the utility of softmax cross-entropy based distillation for re-ranking (Proposition 3, §4.2). We confirm the efficacy of this strategy on the re-reranking benchmarks MSMARCO-Passage (Nguyen et al., 2016) and Natural Questions (Kwiatkowski et al., 2019).

Overall, our results aim to shed light on the fundamental differences between CA and DE models, and give a simple yet effective distillation strategy to boost the performance of the latter.

## 2 Background and notation

**Retrieval problems**. Suppose we have query space $\mathcal{Q}$ and document space $\mathcal{D}$. For a given query $q \in \mathcal{Q}$, the goal of information retrieval is to identify a set of relevant documents $\mathcal{D}_{\mathrm{rel}}(q) \subseteq \mathcal{D}$ (Mitra & Craswell, 2018). This is typically achieved by learning a *scorer* $s \colon \mathcal{Q} \times \mathcal{D} \to \mathbb{R}$ that predicts the *relevance* of a query and document, and simply selecting the top-$k$ highest scoring documents for a given query. Typically, this scorer is itself implemented in two phases (Matveeva et al., 2006; Nogueira & Cho, 2019; Chang et al., 2020; Miech et al., 2021; Hofstätter et al., 2021). In the *retrieval* phase, one identifies an initial candidate set of documents via an initial scorer $s_{\mathrm{ret}}$. In the *re-ranking* phase, one re-scores *only* these candidate documents via a distinct scorer $s_{\mathrm{rrk}}$ to obtain $\mathcal{D}_{\mathrm{rel}}(q)$. We may thus regard $s \colon \mathcal{Q} \times \mathcal{D} \to \mathbb{R}$ as a suitable composition of both $s_{\mathrm{ret}}$ and $s_{\mathrm{rrk}}$.

To learn a scorer, suppose we have supervision $\{(q_n, \mathbf{d}_n, \mathbf{y}_n)\}_{n=1}^N$, where for each query $q_n$ we have a list of $K$ associated documents $\mathbf{d}_n \doteq (d_{n1}, \ldots, d_{nK}) \in \mathcal{D}^K$ with ground-truth relevance labels $\mathbf{y}_n \doteq (y_{n1}, \ldots, y_{nK}) \in \{0,1\}^K$. Typically, $\mathbf{d}_n$ comprises a small subset of $\mathcal{D}$ (i.e., $K \ll |\mathcal{D}|$), since it is rarely feasible to collect relevance labels for *all* documents. Consider a parameterised class of scorers $\mathcal{S} \doteq \{s(\cdot, \cdot; \theta) \colon \mathcal{Q} \times \mathcal{D} \to \mathbb{R} \mid \theta \in \Theta\}$. For any given $\theta \in \Theta$, let $\mathbf{s}_n(\theta) \doteq (s(q_n, d_{n1}; \theta), \ldots, s(q_n, d_{nK}; \theta)) \in \mathbb{R}^K$ denote the vector of model scores on the provided documents. One may then learn a scorer $s$ via minimising $R(\theta) \doteq \frac{1}{N} \sum_{n \in [N]} \ell(\mathbf{y}_n, \mathbf{s}_n(\theta))$ for loss $\ell \colon \mathbb{R}^K \times \mathbb{R}^K \to \mathbb{R}_+$, such as the mean square error $\ell_{\mathrm{mse}}(\mathbf{y}_n, \mathbf{s}_n(\theta)) \doteq \sum_{k \in [K]} (y_{nk} - s_{nk}(\theta))^2$;

and the softmax cross-entropy (assuming for simplicity $\mathbf{1}^\top \mathbf{y} = 1$): for temperature $\tau > 0$,

$$\ell_{\text{ce}}(\mathbf{y}_n, \mathbf{s}_n(\theta)) \doteq -\sum_{k\in[K]} y_{nk} \cdot \log p_{nk}(\theta) \text{ where } p_{nk}(\theta) \doteq \frac{\exp(\tau^{-1} \cdot s_{nk}(\theta))}{\sum_{k'\in[K]} \exp(\tau^{-1} \cdot s_{nk'}(\theta))}. \quad (1)$$

To instantiate either objective, one missing ingredient is the precise parameterisation of $\mathcal{S}$. We now describe one possible parameterisation, given by a transformer model.

**Cross-attention and dual-encoders**. Transformers (Vaswani et al., 2017) are sequence-to-sequence models with good empirical performance on language tasks. Suppose we have a sequence $(x_1, \ldots, x_L)$ of $L$ *tokens* (e.g., words in a language), with $\mathbf{X} \doteq (\mathbf{x}_1, \mathbf{x}_2, \ldots, \mathbf{x}_L) \in \mathbb{R}^{D\times L}$ being a corresponding sequence of token embeddings (e.g., word embeddings). A *transformer encoder* maps this sequence to $\mathbf{Z} \doteq (\mathbf{z}_1, \mathbf{z}_2, \ldots, \mathbf{z}_L) \in \mathbb{R}^{D\times L}$, via a composition of *attention* and *feedforward* layers. The former is a mapping $\texttt{attn}: \mathbb{R}^{D\times L} \to \mathbb{R}^{D\times L}$, which in its simplest ("single head") form is $\texttt{attn}(\mathbf{X}) = \mathbf{X} + \mathbf{W}_V \mathbf{X} \mathbf{A}$, where $\mathbf{A} \doteq \sigma\left(\mathbf{X}^\top \mathbf{W}_Q \mathbf{W}_K \mathbf{X}/\sqrt{D}\right)$ for softmax $\sigma(\cdot)$ and weights $\mathbf{W}_Q, \mathbf{W}_K, \mathbf{W}_V$. Intuitively, this embeds a token via a weighted sum of all tokens' embeddings. The weighting is given by the *attention matrix* $\mathbf{A} \in \mathbb{R}^{L\times L}$, which is a learned similarity measure between pairs of tokens. BERT (Devlin et al., 2019) is a canonical instantiation of the transform encoder, coupled with effective pre-training tasks shown to improve downstream performance in small-sample regimes.

Transformers can readily score (query, document) relevance: given a pair $(q, d) \in \mathcal{Q} \times \mathcal{D}$ of tokenised queries and documents, one may pass their concatenation through a transformer encoder $T$. The resulting embedding may be pooled (e.g., using the [CLS] token (Devlin et al., 2019)) to yield

$$s(q, d) = \mathbf{w}^\top \texttt{pool}(T(\texttt{concat}(q, d)))$$

for weights $\mathbf{w} \in \mathbb{R}^D$. Such *cross-attention* (CA) models apply attention layers on the queries and documents jointly, which intuitively allow for rich interactions.

A distinct strategy is to separately embed queries and documents (e.g., via BERT encoders), and score them with their dot-product (Lee et al., 2019; Chang et al., 2020; Karpukhin et al., 2020; Zhan et al., 2020; Ma et al., 2021; Xiong et al., 2021; Luan et al., 2021; Qu et al., 2021; Zhan et al., 2021a):

$$s(q, d) = \texttt{pool}(T(q))^\top \texttt{pool}(T(d)). \quad (2)$$

Such *dual-encoder* (DE) models have their roots in Siamese networks for image processing (Bromley et al., 1993), and have received recent interest in a range of language problems (Reimers & Gurevych, 2019; Gillick et al., 2019; Karpukhin et al., 2020). A salient feature of Equation 2 is that one can index the query and document embeddings. This implies that DE models can have significantly lower the inference cost compared to CA models, which makes them appealing to use for both the retrieval and re-ranking stages. Indeed, DE models are widely used for retrieval, owing to their amenability to fast approximate nearest neighbour search (Guo et al., 2020; Johnson et al., 2021). Unfortunately, for re-ranking, one typically observes a large gap between DE and CA performance (see §3.2).

**Cross-attention to dual-encoder distillation**. One strategy to bridge the performance gap between CA and DE models is *distillation* (Ba & Caruana, 2014; Hinton et al., 2015). Suppose we have supervision $\{(q_n, \mathbf{d}_n, \mathbf{t}_n, \mathbf{y}_n)\}_{n=1}^N$, where $\mathbf{t}_n \in \mathbb{R}^K$ are the scores from a "teacher" (e.g., CA) model. Then, we may minimise $R_{\text{dist}}(\theta) \doteq \frac{1}{N}\sum_{n\in[N]} \ell(\mathbf{t}_n, \mathbf{s}_n(\theta))$, noting that we use the teacher supervision $\mathbf{t}_n$ rather than "one-hot" labels $\mathbf{y}_n$; one may also combine the teacher and one-hot labels. Two canonical instantiations are *logit matching* (Ba & Caruana, 2014), which uses the the mean square error $\ell_{\text{mse}}(\mathbf{t}_n, \mathbf{s}_n(\theta))$ between the teacher and student scores, and *probability matching* (Hinton et al., 2015), which uses the softmax cross-entropy between the teacher and student softmax probabilities, i.e., $\ell(\mathbf{t}_n, \mathbf{s}_n) = \ell_{\text{ce}}(\mathbf{p}_n^{\text{teach}}, \mathbf{s}_n)$ where $p_{nk}^{\text{teach}} \doteq \frac{\exp(\tau^{-1}\cdot t_{nk})}{\sum_{k'\in[K]}\exp(\tau^{-1}\cdot t_{nk'})}$ for temperature $\tau > 0$. Empirically, distillation has been shown to at least partially bridge the chasm between CA and DE performance (Lu et al., 2020; Izacard & Grave, 2020; Yang & Seo, 2020; Hofstätter et al., 2020a; Miech et al., 2021).

## 3 THE CAPACITY OF DUAL-ENCODER MODELS

We show that while DE models have sufficient capacity to represent complex score functions, in practice they may overfit and thus perform poorly on test data.

### 3.1 How expressive are dual-encoders in theory?

To study the expressivity of DE models, we ask: how well can such models approximate a given score function $s^*\colon \mathcal{Q} \times \mathcal{D} \to \mathbb{R}$? This concerns the theoretical capacity of representing $s^*$, and side-steps the issue of estimating it from samples; we discuss the latter subsequently. For our purposes, the DE family comprises any model of the form $s(q, d) = \mathbf{z}(q)^\top \mathbf{w}(d)$, where $\mathbf{z}\colon \mathcal{Q} \to \mathbb{R}^D$ and $\mathbf{w}\colon \mathcal{D} \to \mathbb{R}^D$ are (arbitrarily powerful) query- and document embedding functions. Under mild assumptions on $\mathcal{D}$, we may show that DE models with sufficiently large embeddings can model a broad class of $s^*$.

**Proposition 1.** *Suppose $\mathcal{D}$ is a compact metric space, and $s^*(q, \cdot)\colon \mathcal{D} \to \mathbb{R}$ is continuous for each $q \in \mathcal{Q}$. Then, for any $\epsilon > 0$, $\exists$ vectors $\mathbf{z}_q, \mathbf{w}_d$ of at most countably infinite dimension such that*

$$(\forall q \in \mathcal{Q}, d \in \mathcal{D}) \, |s^*(q, d) - \mathbf{z}_q^\top \mathbf{w}_d| \le \epsilon.$$

The proof relies on the classic Mercer's theorem for kernel methods, and is provided in the Appendix. We make a few remarks on the above. First, note that by symmetry, we could equally swap the roles of $\mathcal{Q}$ and $\mathcal{D}$ above. Second, the assumption of $\mathcal{D}$ being a compact metric space is mild: we may do this, e.g., by identifying each document with a normalised Euclidean embedding. Third, while the above employs a kernel-based argument, we do *not* require that $s^*$ is symmetric; rather, $s^*(q, \cdot)$ is simply a function to be approximated by an element in a reproducing kernel Hilbert space.

Fourth, one can gain intuition by assuming $\mathcal{Q}$ and $\mathcal{D}$ are finite spaces. Here, we may associate with any $s^*\colon \mathcal{Q} \times \mathcal{D} \to \mathbb{R}$ a matrix $\mathbf{S}^* \in \mathbb{R}^{|\mathcal{Q}| \times |\mathcal{D}|}$, where $S^*_{q,d} \doteq s^*(q, d)$. By the singular value decomposition, $\mathbf{S}^* = \mathbf{Z}\mathbf{W}^\top$ for matrices $\mathbf{Z} \in \mathbb{R}^{|\mathcal{Q}| \times r}, \mathbf{W} \in \mathbb{R}^{|\mathcal{D}| \times r}$, where $r \doteq \mathrm{rank}(\mathbf{S}^*)$. Thus, we may always write $s^*(q, d) = S^*_{q,d} = \mathbf{z}(q)^\top \mathbf{w}(d)$ by simply defining $\mathbf{z}(q) = \mathbf{Z}_{q:}$ and $\mathbf{w}(d) = \mathbf{W}_{:d}$.

Finally, the above does not restrict the function class used to represent either $\mathbf{z}$ or $\mathbf{w}$. Restricting these to transformers, we may appeal to their universal approximation power (Yun et al., 2020): a sufficiently deep transformer encoder can approximate any continuous mapping $T\colon \mathbb{R}^{D \times L} \to \mathbb{R}^{D \times L}$ to arbitrary precision. Now suppose we represent queries $\mathcal{Q} \subset \mathbb{R}^{D \times L}$ as embeddings of sequences of $L$ tokenised elements, with a fixed [CLS] token at the first position. Then, any continuous $\mathbf{z}\colon \mathcal{Q} \to \mathbb{R}^D$ can be approximated by reading the first token embedding of a suitable transformer $T$.

**Proposition 2.** *Fix some compact $\mathcal{Q} \subset \mathbb{R}^{D \times L}$. For any continuous $\mathbf{z}\colon \mathcal{Q} \to \mathbb{R}^D$ and $\epsilon' > 0$, $\exists$ a transformer encoder $T\colon \mathbb{R}^{D \times L} \to \mathbb{R}^{D \times L}$ such that $\|\mathbf{z}(\mathbf{Q}) - T(\mathbf{Q})_{:1}\|_2^2 \le \epsilon'$ for almost every $\mathbf{Q} \in \mathcal{Q}$.*

**Remark**. Luan et al. (2021) similarly analysed the expressive power of CA versus DE models, but through a slightly different lens: they formalised the expressiveness of DE score functions based on *random projections*, showing that the resulting embedding size may need to scale with the document length. This elegant analysis sheds light on one possible failure mode of DE models, i.e., scoring the relevance of queries and overly long documents. By contrast, we do not specifically restrict the form of the score function, which can be more reflective of transformer-based DE models; further, we shall now identify a distinct, fundamental reason for the gap between DE and CA model performance.

### 3.2 How expressive are dual-encoders in practice?

The previous section suggests that, at least in theory, DE models ought to be sufficiently expressive as to model a broad class of scores. However, this is at odds with the wealth of empirical results suggesting a non-trivial performance gap between DE and CA models (Lu et al., 2020; Izacard & Grave, 2020; Hofstätter et al., 2020a; Miech et al., 2021). What is the cause of this discrepancy? Note that the previous section considers only the *capacity* of DE models, i.e., the theoretical existence of a DE model capable of representing a given score function. A distinct question is whether it is feasible to *learn* such a model from a finite number of samples, and with a capacity restriction on the function class (e.g., a fixed embedding dimension and depth of transformer encoder).

To study this, we perform a simple experiment on the benchmark MSMARCO-Passage dataset (Nguyen et al., 2016), which concerns query to passage retrieval; we defer a detailed discussion of this dataset and our training protocol to §5. We focus on the re-ranking setting, wherein both the CA and DE models operate on the outputs of a BM25 retrieval model (Robertson & Zaragoza, 2009). We train a series of BERT-based CA and DE models on the standard ("small") triplets training set, employing 6-layer BERT models from Turc et al. (2019) with varying embedding size. For each such model, we compute the *mean reciprocal rank* (MRR)@10 (Radev et al., 2002) on the provided train and dev set.

Figure 2: Comparison of CA (left) and DE (middle) model predictions on the MSMARCO-passage test set. The CA model more confidently distinguishes positives from negatives, with a pronounced peak in scores for the latter. For visual clarity, the scores are translated to have mean zero; see Figure 6 for an uncentered plot. We further see that the *margins* of the DE model are distinctly smaller than CA (right); this may be mitigated with suitable distillation (DIST).

Figure 1 compares the train and test MRR@10 for cross-attention (CA) and dual-encoder (DE) models. See Table 2 (Appendix) for a numeric summary. With increased embedding dimension, the DE model closely matches the performance of the CA on the *training* set. This is consistent with the previous section, which suggests suitably deep transformer-based DE models ought to model any ground truth scores; however, there is a sizable gap on the *test* set. This points to the DE models suffering from *generalisation*, rather than *capacity*. See Appendix C.1 for similar results with a 2-layer BERT model.

## 3.3 What causes the dual-encoder generalisation gap?

We now attempt to further understand the cause for the DE model's generalisation gap. We perform a more fine-grained inspection of the CA versus DE model predictions by studying the predicted scores for positive and negative pairs. More precisely, for each query $q$ in the test set, we consider the distribution of scores for positive and negative (query, document) pairs $(q, d^+)$ and $(q, d^-)$.

Figure 2 shows that both CA and DE models possess clear modes for the positive and negative pairs. However, the CA model can be seen to not "waste" its capacity by unnecessarily modelling fine-grained distinctions amongst negative pairs; rather, it collapses most negatives to a small range of scores, and focusses instead on clearly separating the positives and negatives. By contrast, the DE model has far more diffuse negative scores. This results in a greater overlap in the positive and negative scores, implying the model has more difficulty making distinctions between these pairs. From the right plot, we also see that the DE model has smaller *normalised margins* $(s(q, d^+) - s(q, d^-))/\rho$ between the positive and negative pairs, where $\rho$ is the maximal score range.

The DE score distribution for negatives is plausible: when updating a DE model for a given $(q, d)$ pair, the factorised nature of the model implies a non-trivial influence on the scores for *all* other pairs $(q, d')$ and $(q', d)$ sharing either the query or document. Consequently, it is intuitively harder for the model to arrive at a solution that makes fine-grained distinctions. This can be verified qualitatively; the DE model is often seen to make errors on pairs which have only superficial relevance (Appendix C.4).

We make a few remarks here. First, similar trends to the above holds on the training set; see Figure 5 (Appendix). Second, for visual clarity, in Figure 2 we apply a constant offset to each model's scores to ensure they are mean zero. Figure 6 (Appendix) provides an uncentered plot, showing the DE model scores to have a strong *baseline shift* over the CA model.

## 3.4 How can we mitigate the dual-encoder generalisation gap?

Having identified the overfitting problem plaguing DE models, one may naturally ask how to mitigate this. One natural option is to apply standard regularisation strategies, such as dropout; however, our experiments reveal these to be largely ineffective in mitigating DE overfitting (see Appendix C.10). Alternately, the recent literature has identified two promising (complementary) strategies. The first is to modify the *scoring function* used to compute the final DE model score based on the query and document embeddings; i.e., we replace Equation 2 with $s(q, d) = \texttt{score}(T(q), T(d))$ for some scoring function score (see, e.g., MacAvaney et al., 2020; Khattab & Zaharia, 2020; Luan et al., 2021). Such strategies have proven effective; however, as these involve some overhead over dot-product scoring, it is natural to ask whether one can extract more gains out of the latter. The second strategy is

to *distill* the predictions of the CA model into the DE model. By itself, this does not require changing the dot-product scoring in Equation 2. While distillation has been widely studied for classification settings, its use for re-ranking settings is less well-developed. We now study this issue more closely.

## 4    IMPROVING DUAL-ENCODERS VIA DISTILLATION

We now propose a family of distillation schemes to improve the performance of DE models.

### 4.1    FROM LOGIT TO MARGIN MATCHING

Recall that in distillation settings, each sample is the form $(q, \mathbf{d}, \mathbf{y}, \mathbf{t})$, where for query $q$ and documents $\mathbf{d} \in \mathcal{D}^K$, we have ground-truth relevance labels $\mathbf{y} \in \{0, 1\}^K$ and "teacher" model scores $\mathbf{t} \in \mathbb{R}^K$ (e.g., the outputs of a CA model). To fit scores $\mathbf{s} \in \mathbb{R}^K$ for a "student" model (e.g., a DE model), one can employ a suitable loss $\ell(\mathbf{t}, \mathbf{s})$. Per §2, the simplest choice is the *logit matching* or *mean square error* loss. Let us momentarily assume that for each query we have a single associated positive and negative document, with respective teacher scores $\mathbf{t} = (t_1, t_2)$ and student scores $\mathbf{s} = (s_1, s_2)$. Then, the loss simply minimises $\ell_{\mathrm{mse}}(\mathbf{t}, \mathbf{s}) = \|\mathbf{t} - \mathbf{s}\|_2^2 = (t_1 - s_1)^2 + (t_2 - s_2)^2$.

While intuitive, this loss makes the implicit assumption that the range of scores for teacher and student are commensurate. When this is violated — as we empirically observe with the *baseline shift* between CA and DE model scores, per discussion in §3.2 and Figure 6 (Appendix) — the loss may be ineffective. To get around this, the margin MSE loss (Hofstätter et al., 2020a) matches *margins*, via

$$\ell_{\mathrm{mmse}}(\mathbf{t}, \mathbf{s}) \doteq ((t_1 - t_2) - (s_1 - s_2))^2. \tag{3}$$

Clearly, this loss is translation invariant, and thus is immune to baseline shifts in the scores. However, two issues remain. First, the loss is tied to triplet data with a single positive and negative document. Intuitively, it ought to be beneficial to leverage the teacher scores for *multiple* documents, with $\mathbf{t}, \mathbf{s} \in \mathbb{R}^K$ for generic $K$. One natural means of achieving this is $\ell(\mathbf{t}, \mathbf{s}) = \sum_{i \in P} \sum_{j \in N} ((t_i - t_j) - (s_i - s_j))^2$, where $P, N$ denote the set of positive and negative documents, with $P \cup N = [K]$.

Second, the above loss demands matching *all* documents' margins *exactly*. However, this may be an ineffective use of model capacity: for re-ranking tasks, it is primarily important to ensure a clear *separation* between the positive and negative documents, and overcome the overlapping scores observed in Figure 2. To this end, suppose then that the negative documents are sorted in descending order of the teacher scores $\mathbf{t} = (t_1, \ldots, t_K)$. We propose to extend Equation 3 via

$$\ell_{\mathrm{m3se}}(\mathbf{t}, \mathbf{s}) = \sum_{i \in P} ((t_i - t_{j^*}) - (s_i - s_{j^*}))^2 + \sum_{j \in N} [s_j - s_{j^*}]_+^2, \tag{4}$$

where $j^* \in N$ denotes the index of the negative document with highest teacher score. This *multi-margin MSE* ($M^3SE$) loss only attempts to match the margins for the highest scoring negative $j^*$; for all other negatives $j \neq j^*$, we simply encourage them to have lower score than $j^*$. While one may design other intuitively plausible losses, an appealing property of Equation 4 is its relation to the recently proposed RankDistil framework (Reddi et al., 2021), as shall be discussed in §4.3.

### 4.2    FROM MARGIN TO PROBABILITY MATCHING

The above extended the logit matching distillation loss for re-ranking settings, with care to account for differing score ranges for teacher and student. Per §2, the other classic distillation loss is the softmax cross-entropy (Equation 1), which has seen prior use for dual-encoder distillation (Lu et al., 2020; Izacard & Grave, 2020; Yang & Seo, 2020; Hofstätter et al., 2020a; Miech et al., 2021). In fact, we now show that the softmax cross-entropy generalises the margin MSE loss in a precise sense.

**Proposition 3.** *Fix any teacher and student scores* $\mathbf{t}, \mathbf{s} \in \mathbb{R}^2$. *Let* $\ell_{\mathrm{sce}}(\mathbf{t}, \mathbf{s}; \tau)$ *denote the softmax cross-entropy loss (Equation 1) with temperature* $\tau > 0$, *and* $\ell_{\mathrm{mmse}}$ *the margin MSE loss (Equation 3). Then, for temperature scaled sigmoid function* $\sigma(z; \tau) \doteq (1 + \exp(-\tau^{-1} \cdot z))^{-1}$, *we have*

$$\ell_{\mathrm{sce}}(\mathbf{t}, \mathbf{s}; \tau) = \mathrm{KL}_{\mathrm{bin}}(\sigma(t_1 - t_2; \tau) \, \| \, \sigma(s_1 - s_2; \tau))$$

$$\lim_{\tau \to +\infty} \left[ \tau^2 \cdot \frac{\partial}{\partial s_i} \ell_{\mathrm{sce}}(\mathbf{t}, \mathbf{s}; \tau) \right] = \frac{1}{8} \cdot \frac{\partial}{\partial s_i} \ell_{\mathrm{mmse}}(\mathbf{t}, \mathbf{s}) \, \textit{for } i \in \{1, 2\}.$$

We make a few remarks. First, $\ell_{\mathrm{sce}}$ swaps the square loss for a binary KL divergence $\mathrm{KL}_{\mathrm{bin}}(a\|b) \doteq -a \cdot \log \frac{b}{a} + (1-a) \cdot \log \frac{1-b}{1-a}$ in Equation 3. Unlike the square loss, the softmax cross-entropy is Lipschitz, which is typically favourable for generalisation (Luxburg & Bousquet, 2004). Second, the softmax and margin MSE loss derivatives with respect to the student logits converge in the high temperature limit. This gives a precise sense in which the softmax cross-entropy is a *smooth approximation* to the margin MSE. Intuitively, this is plausible: the former is similarly invariant to constant translations to the scores, thus allowing for differing ranges in the teacher and student.

Third, our argument is subtly different from Hinton et al. (2015), which establishes convergence of the softmax and *logit matching* derivatives: crucially, the latter only holds under the assumption that the teacher and student logits are centered. We reiterate that this assumption typically does *not* hold for CA and DE models, per §3.2 and Figure 6 (Appendix). (While it is benign to translate all teacher logits by a fixed constant, this may not guarantee the assumption as the average score $\frac{1}{K} \sum_{i \in [K]} t_i$ is typically *not* constant across samples.) By contrast, Proposition 3 makes no such assumption.

Finally, Proposition 3 assumes the supervision comprises a a single positive and negative document, to facilitate comparison with Equation 3. The case of $K > 2$ documents with teacher and student scores $\mathbf{t}, \mathbf{s} \in \mathbb{R}^K$ poses no conceptual issue: we may seamlessly minimise the softmax cross-entropy between the two vectors. Since typically $K \ll |\mathcal{D}|$, this does *not* correspond to matching the full teacher distribution over all possible "classes". Despite this distinction to the standard use of distillation for classification problems, one may justify the loss as follows. For positive (negative) documents $P$ ($Q$),

$$\ell_{\mathrm{sce}}(\mathbf{t}, \mathbf{s}) \propto \frac{1}{\tau} \sum_{k \in [K]} \frac{e^{\tau^{-1} \cdot t_k}}{\sum_{i \in P} e^{\tau^{-1} \cdot t_i} + \sum_{j \in N} e^{\tau^{-1} \cdot t_j}} \cdot \log \left[ \sum_{i \in P} e^{\tau^{-1} \cdot (s_i - s_k)} + \sum_{j \in N} e^{\tau^{-1} \cdot (s_j - s_k)} \right].$$

This approaches $\max \{\max_{i \in P}[s_i - s_{k^*}]_+, \max_{j \in N}[s_j - s_{k^*}]_+\}$ as $\tau \to +\infty$, where $k^*$ is the highest scoring document under the teacher. The loss thus encourages the student to preserve the teacher's highest scoring document, which is useful for re-ranking. Compared to Equation 4, this does not explicitly delineate the highest scoring negative from other negatives; however, empirically, we shall see that the departure from square loss endows the loss with comparable efficacy.

### 4.3 DISCUSSION AND EXTENSIONS

**Connection to RankDistil**. Equation 4 can be related to the recently proposed RankDistil framework by Reddi et al. (2021). Here, the goal is to design distillation losses to match the teacher and student *ranking* over labels, particularly for the top-$k$ elements. A canonical instantiation is the *binary* RankDistil formulation (RankDistil-B), $\ell_{\mathrm{rankB}}(\mathbf{t}, \mathbf{s}) = \sum_{i \in P}(t_i - s_i)^2 + \sum_{j \in N}[s_j - \gamma_0]_+^2$. Intuitively, this seeks to match the teacher and student score on positive documents, while ensuring all negative documents fall below some threshold $\gamma_0$. The relation between this loss and Equation 4 is analogous to that between logit matching and margin MSE: Equation 4 matches *margins* rather than raw scores, which are thus suitable for settings of differing ranges of the teacher and student scores.

**Distillation for retrieval phase**. Our focus thus far has been on bridging the observed gap between the CA and DE models for re-ranking. However, the losses presented above are equally applicable during the retrieval phase, where they can similarly be expected to improve DE model performance. These losses are *complementary* to proposals from several recent works improving the retrieval performance of dual-encoder models, including using suitable hard negative mining for training (Xiong et al., 2021; Zhan et al., 2021a), and quantisation (Zhan et al., 2021b).

## 5 EXPERIMENTAL RESULTS

We now present experiments that demonstrate the viability of the proposed distillation scheme in §4. We show that this scheme yields better performance compared to several baselines, helping reduce the gap between CA and DE models by mitigating the overfitting issue identified in §3.2.

Per §4.3, we reiterate that our primary goal is to explore the feasibility of bridging the gap between CA and DE models for *re-ranking*. This is distinct from the more common use of DE models in the literature for *retrieval* (Chang et al., 2020; Xiong et al., 2021; Zhan et al., 2021a). While we do not make special effort to incorporate tricks from that literature that are beneficial for retrieval

| Model | MSMARCO re-rank | | MSMARCO retrieval | | TREC DL19 re-rank | | NQ re-rank | |
|---|---|---|---|---|---|---|---|---|
| | MRR | nDCG | MRR | nDCG | MRR | nDCG | MRR | nDCG |
| **Baselines: one-hot** | | | | | | | | |
| BM25 (Robertson & Zaragoza, 2009) | 0.194† | 0.241† | 0.194† | 0.241† | 0.689† | 0.501† | — | — |
| ANCE (Xiong et al., 2021) | — | — | 0.330† | — | — | 0.677† | — | — |
| Cross-attention BERT (12-layer) | 0.370 | 0.430 | N/A | N/A | 0.829 | 0.749 | 0.746 | 0.673 |
| Dual-encoder BERT (6-layer) | 0.310 | 0.360 | 0.281 | 0.331 | 0.834 | 0.677 | 0.676 | 0.601 |
| **Distilled dual-encoder: prior work** | | | | | | | | |
| MSE (Hofstätter et al., 2020a) | 0.289 | 0.343 | 0.000* | 0.000* | 0.000* | 0.000* | 0.659 | 0.591 |
| Margin MSE (Hofstätter et al., 2020a) | 0.334 | 0.392 | 0.319 | 0.375 | 0.867◇ | 0.718 | 0.673 | 0.594 |
| RankDistil-B (Reddi et al., 2021) | 0.249 | 0.301 | 0.000* | 0.000* | 0.000* | 0.000* | 0.649 | 0.561 |
| **Distilled dual-encoder: this work** | | | | | | | | |
| M³SE (Equation 4) | 0.349 | 0.406 | 0.337 | 0.394 | 0.852 | 0.714 | 0.699 | 0.625 |
| Softmax CE (Equation 1) | 0.346 | 0.405 | 0.334 | 0.392 | 0.846 | 0.726◇ | 0.682 | 0.607 |

Table 1: Summary of MRR@10 and nDCG@10 for all methods on MSMARCO Passage and Natural Questions (NQ). We compare cross-attention, dual-encoder, and distilled dual-encoder BERT models. We highlight the best performing DE based model. Distilling the dual-encoder with our proposed techniques significantly improves performance over one-hot training and existing distillation techniques. Results marked † are quoted from the corresponding reference, "N/A" are not applicable (e.g., the cross-attention model is not feasible to apply for retrieval), and "—" are not available from the reference. ◇ See Appendix C.11 for analysis of potential label noise influencing the results. * See text for discussion.

(e.g., using hard negatives), we do demonstrate that our proposed distillation techniques can improve performance for this stage over simply using one-hot training labels.

**Datasets**. We present results on MSMARCO-Passage (Nguyen et al., 2016) and Natural Questions (NQ) (Kwiatkowski et al., 2019). For MSMARCO, we report results on the standard dev set, and the TREC DL19 test set (Craswell et al., 2020). MSMARCO comprises (query, passage) records and their relevance labels from a human rater; for each query, the associated passages are retrieved from a BM25 model. The training set is canonically represented in *triplet* form, comprising a query, and a single positive and negative passage. NQ comprises user questions and Wikipedia passages potentially containing their answers; we use a processed version from Karpukhin et al. (2020) (cf. Appendix B).

**Models**. We use transformer encoders initialised with the standard pre-trained BERT model checkpoints. Following Hofstätter et al. (2020a), we use BERT-Base for the CA model, and a 6-layer BERT model (Turc et al., 2019) with embedding size 768 for all DE models. For the DE models, we tie the query and document encoder parameters. For distillation, we use the CA model as the "teacher" and the DE model as the "student". On MSMARCO, for ease of comparison, we use the "T1" CA model annotations from Hofstätter et al. (2020a), which achieves similar re-re-ranking performance as our independently trained CA model. See Appendix B for further training details.

**Metrics**. We report the MRR@10 and nDCG@10 for all methods. Note that we are primarily interested in *re-ranking* tasks, wherein the test set comprises documents retrieved by a BM25 baseline, and our models are used to re-rank these candidates. For MSMARCO, we additionally consider full retrieval performance obtained by scoring *all* (8.8M) passages, but reiterate that maximising performance on this metric requires additional negative mining that we do not explore.

**Baselines**. We consider methods that leverage either the observed ("one-hot") training labels on the triplet data, or the predictions from the cross-attention ("teacher") model on the triplet data. For the former, we employ softmax cross-entropy against the using a cross-attention and dual-encoder model. For the latter, we employ the logit MSE loss (Ba & Caruana, 2014), margin MSE loss (Hofstätter et al., 2020a), and the binary version of RankDistil (RankDistil-B) (Reddi et al., 2021).

These are compared against the proposed M³SE (Equation 4), and the softmax cross-entropy (Equation 1). The latter methods handle of an arbitrary number of documents $K$ in the supervision (i.e., they are not restricted to triplet data). We thus aggregate the triplet data by query, and retain the top-20 passages with highest teacher model score. We study the sensitivity to $K$ in Appendix C.7.

## 5.1 Results and discussion

Table 1 summarises the results of all methods. We discuss a few salient findings.

**Baseline performance**. Comparing the performance of CA and DE models in Table 1, we make two initial observations. First, there is a sizable performance gap between the models when using one-hot labels. Second, in line with Hofstätter et al. (2020a), naïve logit matching via the MSE underperforms, due to the vastly different scales of the two models; this is however assuaged by the margin MSE loss.

**Efficacy of distillation**. Both the $M^3SE$ and the softmax cross-entropy losses significantly outperform existing methods on the MSMARCO and NQ re-reranking tasks, and shrinks the gap between the DE and CA models. On the TREC DL19 test set, our proposed methods appear to fall short of the margin MSE MRR@10; a closer inspection of the apparent loss cases (Appendix C.11) reveals these are unanimously the results of false negatives in the provided labels.

Figure 1 shows that $M^3SE$ distillation nearly closes the gap between CA and DE models with sufficiently large embedding dimension. Intriguingly, the *test* gains comes with some sacrifice of *training* MRR for the highest embedding size. This is consistent with findings in the classical distillation literature, where distillation can harm training accuracy but improve test accuracy (Cho & Hariharan, 2019). To further study this, Figure 4 (Appendix) shows the learning curve on the MSMARCO train and test set, which demonstrates a consistent gap in test set performance as training progresses. Distillation yields a training MRR that closely matches that of the DE model for the first $100K$ training steps; subsequently, however, the training MRR under distillation begins to saturate, while that of the DE model with one-hot labels keeps increasing. Intuitively, this suggests that the DE model starts exploring a poorly generalising part of the parameter space beyond this point.

As a final illustration, Figure 2 (right) shows that softmax cross-entropy distillation makes the DE scores better behaved: it shifts the margins of the DE model to be closer to the the CA model.

**Beyond the dot-product: generic scoring functions**. We have thus far focussed on results using the standard dot-product based scoring for the DE models, per Equation 2. It is of interest as to what impact more complex scoring functions, per §3.4, have on final model performance. To study this, we consider the ColBERT model (Khattab & Zaharia, 2020), which computes the average of the maximum query-document token similarities. Consistent with Khattab & Zaharia (2020), using this model with one-hot labels by itself improves performance significantly over the dot-product.

When further combined with distillation, the performance *exceeds that of the CA teacher*, reaching an MRR@10 of **0.376** for the MSMARCO re-ranking task under the softmax cross-entropy loss. (Full results in Appendix, Table 7.) This further highlights that factorised representations need not imply a loss in performance, provided they are suitably trained. We defer to future work a similar analysis of the conceptually similar multi-vector scoring layer of Luan et al. (2021).

**Full-retrieval performance**. Thus far, we have focussed on the re-ranking performance of all models, wherein the predictions from a BM25 retriever are provided as input. This allows for the comparison of DE and CA models on an equal footing. One may however naturally wonder how the proposed methods fare in the retrieval phase, wherein the models score *all* possible passages. Table 1 also summarises the results in this setting on MSMARCO. We again see that the proposed distillation techniques offer strong gains over standard one-hot training of the DE model, as well as existing distillation techniques. We remark also that the poor retrieval performance of the MSE loss is a result of the model strongly overfitting to the teacher scores on the provided documents.

**Additional results**. In the Appendix, we present several additional results and ablations, including the sensitivity of performance to the number of documents $K$ in the distilled supervision (Table 6); the effectiveness of label smoothing as opposed to full distillation (Appendix C.9); and the impact of other regularisation strategies aimed at minimising overfitting (Appendix C.10).

## 6 CONCLUSION AND FUTURE WORK

Several avenues for future work remain open. For example, in the literature on distilling between cross-attention models, a common strategy is to match more than simply output logits (Sun et al., 2019; Turc et al., 2019; Sanh et al., 2019; Xu et al., 2020; Sun et al., 2020; Jiao et al., 2020; Wang et al., 2020). Are such strategies useful when distilling to a dual encoder? Further, our distillation strategy assumes access to scores from a cross-attention model. Can one do away with this altogether, e.g., using self-distillation and related ideas (Furlanello et al., 2018)? Finally, the efficacy of the proposed losses when combined with recent advances for retrieval (Xiong et al., 2021) is of interest.

ETHICS STATEMENT

This work concerns the mathematical and empirical analysis of techniques for neural re-ranking, a particular problem encountered in information retrieval. We do not foresee the former having undue societal effects. The latter does not explicitly consider issues of fairness in retrieval, which is an important yet under-studied dimension. We do not foresee our distillation techniques as unduly amplifying biases in existing algorithms; nonetheless, we consider a more careful study of the topic of fairness in retrieval an important avenue for future work.

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

## A PROOFS

*Proof of Proposition 1.* For any $q \in \mathcal{Q}$, let us write $s_q \doteq s(q, \cdot) \colon \mathcal{D} \to \mathbb{R}$. Pick a reproducing kernel Hilbert space $\mathcal{H}$ over $\mathcal{D}$ with associated inner-product $\langle \cdot, \cdot \rangle_{\mathcal{H}}$. Suppose further $\mathcal{H}$ has associated kernel $k \colon \mathcal{D} \times \mathcal{D} \to \mathbb{R}$ that is continuous, and universal (Micchelli et al., 2006). Then, $\mathcal{H}$ is dense in the space of continuous functions (Sriperumbudur et al., 2011). Recall that by assumption $s_q$ is continuous. Consequently, each $s_q$ can be approximated to arbitrary precision (with respect to the $\ell_\infty$ metric) by some $s_q^* \in \mathcal{H}$. Since $s_q^* \in \mathcal{H}$, by the reproducing property of an RKHS,

$$(\forall d \in \mathcal{D}) \, s^*(q, d) = \langle s_q^*, k_d \rangle_{\mathcal{H}},$$

where in an abuse of notation $k_d \doteq k(d, \cdot) \in \mathcal{H}$. Observe that we may interpret $s_q^*$ as an "embedding" for the query $q$, and $k_d$ a "weight vector" for the document $d$.

It may appear at this stage that we are done; however, each of $s_q^*$ and $k_d$ may be uncountably infinite dimensional objects, and the inner product in $\mathcal{H}$ may not be the standard dot-product. Nonetheless, under the stated assumptions (in particular, compactness of $\mathcal{D}$), we may appeal to a Mercer representation of the kernel, following Steinwart & Christmann (2008, Theorem 4.51). In particular, we may construct a *Mercer feature map* $\Phi \colon \mathcal{D} \to \ell_2$ for $\mathcal{H}$, such that

$$(\forall h \in \mathcal{H}) \, (\exists (a_n)_{n=1}^\infty \in \ell_2) \, h(d) = \sum_{n=1}^\infty a_n \cdot \Phi_n(d).$$

Further, the inner product between any $h_1, h_2 \in \mathcal{H}$ with "Mercer coefficients" $(a_n)_{n=1}^\infty, (b_n)_{n=1}^\infty$ is expressible as

$$\langle h_1, h_2 \rangle_{\mathcal{H}} = \sum_{n=1}^\infty a_n \cdot b_n,$$

i.e., a familiar dot-product. In our context, write the Mercer coefficients for $s_q^*$ as $(z_{qn})_{n=1}^\infty$, and for $k_d$ as $(w_{dn})_{n=1}^\infty$. Thus, we may write

$$\langle s_q^*, k_d \rangle_{\mathcal{H}} = \mathbf{z}_q^\top \mathbf{w}_d,$$

where the vectors $\mathbf{z}_q, \mathbf{w}_d$ have at most countably infinite dimension.

$\square$

*Proof of Proposition 2.* Any continuous $\mathbf{z} \colon \mathbb{R}^{D \times L} \to \mathbb{R}^D$ can be trivially associated with $\tilde{\mathbf{z}} \colon \mathbb{R}^{D \times L} \to \mathbb{R}^{D \times L}$, where each column $\tilde{\mathbf{z}}_{:i} = \mathbf{z}$. Such a mapping preserves continuity. By Yun et al. (2020, Theorem 2), we may thus find a transformer encoder $\mathbf{T} \colon \mathbb{R}^{D \times L} \to \mathbb{R}^{D \times L}$ such that $\int_{\mathcal{Q}} \|\tilde{\mathbf{z}}(\mathbf{Q}) - \mathbf{T}(\mathbf{Q})\|_2^2 \, d\mathbf{Q} \leq \epsilon'$. Since the integrad is nonnegative, we must also have $\|\tilde{\mathbf{z}}(\mathbf{Q}) - \mathbf{T}(\mathbf{Q})\|_2^2 \, d\mathbf{Q} \leq \epsilon'$ for any given $\mathbf{Q}$. Consequently, we must also have $\|\tilde{\mathbf{z}}(\mathbf{Q})_{:1} - \mathbf{T}(\mathbf{Q})_{:1}\|_2^2 \leq \epsilon'$ almost surely, which by definition implies $\|\mathbf{z}(\mathbf{Q}) - \mathbf{T}(\mathbf{Q})_{:1}\|_2^2 \leq \epsilon'$, $\square$

*Proof of Proposition 3.* For temperature $\tau > 0$, simple algebra reveals Equation 1 to be

$$\ell_{\text{sce}}(\mathbf{t}, \mathbf{s}) = - \sum_{y \in \{1,2\}} \frac{\exp(\tau^{-1} \cdot t_y)}{\exp(\tau^{-1} \cdot t_1) + \exp(\tau^{-1} \cdot t_2)} \cdot \log \frac{\exp(\tau^{-1} \cdot s_y)}{\exp(\tau^{-1} \cdot s_1) + \exp(\tau^{-1} \cdot s_2)}$$

$$= \sum_{z \in \{\pm 1\}} \frac{1}{1 + e^{z \cdot \tau^{-1} \cdot (t_2 - t_1)}} \cdot \log(1 + e^{z \cdot \tau^{-1} \cdot (s_2 - s_1)})$$

$$= - \sum_{z \in \{\pm 1\}} \sigma(z \cdot (t_1 - t_2); \tau) \cdot \log(\sigma(z \cdot (s_1 - s_2); \tau)).$$

This is equivalent to $\text{KL}_{\text{bin}}(\sigma(t_1 - t_2; \tau) \,\|\, \sigma(s_1 - s_2; \tau))$, i.e., swapping out the square loss for the KL divergence in Equation 3. In fact, one can make a stronger connection: observe that

$$\frac{\partial}{\partial s_1} \ell_{\text{mmse}}(\mathbf{t}, \mathbf{s}) = 2 \cdot ((s_1 - s_2) - (t_1 - t_2))$$

$$\frac{\partial}{\partial s_1} \ell_{\mathrm{sce}}(\mathbf{t}, \mathbf{s}) = - \sum_{z \in \{\pm 1\}} \sigma(z \cdot (t_1 - t_2); \tau) \cdot \frac{\partial}{\partial s_1} \log(\sigma(z \cdot (s_1 - s_2); \tau)$$

$$= \tau^{-1} \cdot \left( \frac{e^{\tau^{-1} \cdot s_1}}{e^{\tau^{-1} \cdot s_1} + e^{\tau^{-1} \cdot s_2}} - \frac{e^{\tau^{-1} \cdot t_1}}{e^{\tau^{-1} \cdot t_1} + e^{\tau^{-1} \cdot t_2}} \right)$$

$$= \tau^{-1} \cdot \left( \frac{1}{1 + e^{\tau^{-1} \cdot (s_2 - s_1)}} - \frac{1}{1 + e^{\tau^{-1} \cdot (t_2 - t_1)}} \right)$$

$$\sim \tau^{-2} \cdot \left( \frac{s_1 - s_2}{4} - \frac{t_1 - t_2}{4} \right),$$

as $\tau \to +\infty$, where the last line follows from the Taylor series approximation

$$\frac{1}{1 + \exp(-z)} = \frac{1}{2} + \frac{z}{4} + \mathcal{O}(z^2)$$

as $z \to 0$. $\qquad\square$

## B    EXPERIMENT HYPERPARAMETERS

For all models, at the output layer we apply dropout at rate $0.1$ and layer normalisation. We use a sequence length of 30 for queries, and 200 for passages. We optimise all methods for a maximum of $3 \times 10^5$ steps using Adam with weight decay, with a batch size of 128 and a learning rate of $2.8 \times 10^{-5}$ (i.e., a $4\times$ scaling of the choices in Hofstätter et al. (2020a)). Following Hofstätter et al. (2020a), we perform early stopping based on the nDCG@10 metric.

For the Natural Questions dataset, we use the processed version used in Karpukhin et al. (2020), that contains questions, positive passages with a correct answer to each question, and a corpus of all Wikipedia passages. To train DE, CA, and multi-negative DIST models, we use 19 BM-25 hard-negative passages for each question along with a positive passage. For single-negative DIST models we use a single BM-25 negative from the same collection. To calculate MRR@10 and nDCG@10 metrics, we use the queries in the dev set with 200 passages containing positives, 100 BM-25 hard-negatives and up to 100 random negatives.

## C    ADDITIONAL EXPERIMENTS

### C.1    CA VERSUS DE MODELS: EFFECT OF BASE ARCHITECTURE

Figure 3 presents an an analogue of Figure 1, but using instead a 2-layer BERT model. As before, we initialise using the pre-trained models developed in Turc et al. (2019). The same general trends hold: for sufficiently large embedding dimension, the DE model closely matches the performance of the CA on the *training* set; however, there is a sizable gap in the *test* set performance. Table 2 presents a numeric summary of the train and test performance for CA and DE models, under different choices of base BERT-model architecture (in terms of number of layers, as well as the final embedding dimension).

### C.2    CA VERSUS DE MODELS: EVOLUTION OF TRAIN AND TEST PERFORMANCE

The above results summarise the results at the completion of training. For a finer-grained understanding of how performance evolves during training, Figure 4 presents the learning curves for CA, DE, and distilled DE (DIST) models on the train and test set. Here, we use the small-bert-6-768 architecture for all models. We observe that the CA and DE models initially have a non-trivial gap in performance, but this shrinks over time; this further points to the initialisation of the CA models being more favourable for generalisation. Note also that beyond a certain point, both models continually improve their *training* performance, but are completely neutral in terms of *test* performance. Interestingly, distillation largely tracks the DE model performance, but then *worsens* on the training set while *improving* on the test set. This suggests that distillation helps explore a better part of the search space.

| Model | MRR@10 train | | MRR@10 test | |
| --- | --- | --- | --- | --- |
| | CA | DE | CA | DE |
| small-bert-2-128 | 0.356 | 0.292 | 0.309 | 0.221 |
| small-bert-2-256 | 0.358 | 0.302 | 0.322 | 0.253 |
| small-bert-2-512 | 0.398 | 0.374 | 0.335 | 0.268 |
| small-bert-2-768 | 0.429 | 0.406 | 0.334 | 0.281 |
| small-bert-6-128 | 0.385 | 0.317 | 0.334 | 0.249 |
| small-bert-6-256 | 0.427 | 0.407 | 0.350 | 0.282 |
| small-bert-6-512 | 0.493 | 0.491 | 0.360 | 0.293 |
| small-bert-6-768 | 0.522 | 0.518 | 0.364 | 0.310 |
| bert-base | 0.658 | 0.635 | 0.368 | 0.309 |

Table 2: Comparison of cross-attention (CA) and dot-product based dual-encoder (DE) models on the MSMARCO-Passage re-ranking task. Notably, for sufficiently large embedding dimension, the DE model closely matches the performance of the CA on the *training* set; however, there is a sizable gap on the *test* set. This points to the poorer DE model performance being largely an issue of generalisation, rather than capacity.

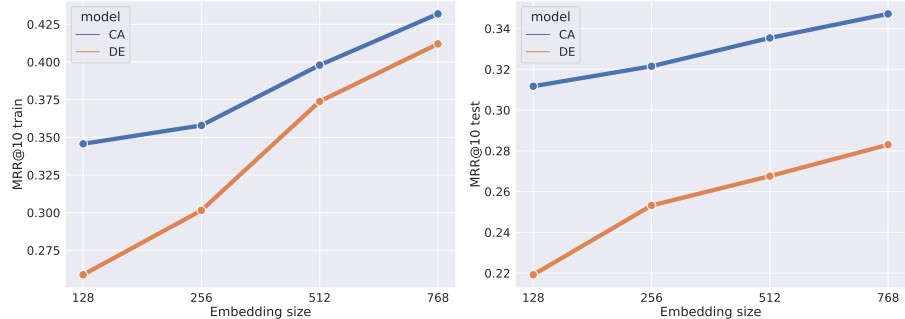

Figure 3: Comparison of BERT-based cross-attention (CA) and dual-encoder (DE) models on the MSMARCO-Passage re-ranking task. For varying embedding dimension of an underlying 2-layer BERT model, we report the train and dev set MRR@10. We see the same trend as in Figure 1: for sufficiently large embedding dimension, the DE model closely matches the performance of the CA on the *training* set; however, there is a sizable gap in the *test* set performance.

## C.3 CA versus DE models: comparison of score distributions

Figure 5 compares the score distributions for the CA and DE models on the *training set*. We observe the same general trends as the test set (Figure 2). Figure 6 further presents the test score distributions without zero-centering. We see that the DE model exhibits a strong baseline shift, wherein the scores are of the order of $\sim 700$.

Figure 7 studies the score distributions for the CA and DE models after varying numbers of training steps. At initialisation, as expected, both models have essentially overlapping distributions for the positives and negatives. However, here too, we observe that the DE model scores operate on a much wider range than the DE model. The middle row continues this study by looking at behaviour after $10,000$ steps of training, which is a small fraction of the $300,000$ steps used for training of all models. Here, we start to observe a clearer separation between the positive and negative scores for both models. Interestingly, the CA model already sees a sharp peak on the negatives, unlike the DE model. The bottom row shows the test set score distributions of the final trained models, to complement Figure 2 from the body. We see the same general trend as the training set: the CA model is seen to more confidently distinguish positives from negatives, and operate at a narrower range of scores.

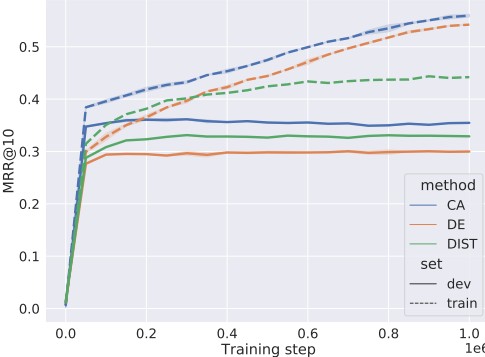

Figure 4: Learning curve for CA, DE, and DIST models on the MSMARCO-Passage train and test set. Here, we use the small-bert-6-768 architecture. Distillation is seen to saturate training performance beyond a certain point, while still resulting in a solution with better generalisation on the test set.

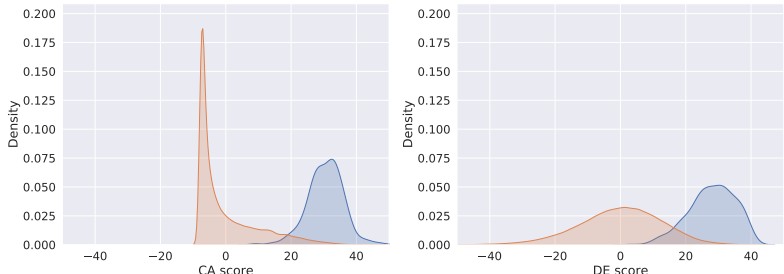

Figure 5: Comparison of CA and DE model predictions on the MSMARCO-Passage train set. We observe the same general trends as the test set (Figure 2). For visual clarity, the scores are translated to have mean zero.

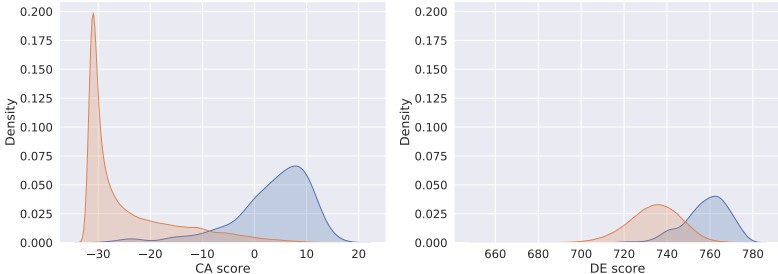

Figure 6: Comparison of CA and DE model predictions on the MSMARCO-Passage test set. Here, we do *not* center the scores to have mean zero. We see that the DE model scores possess a strong baseline shift over the model scores.

## C.4  CA versus DE models: qualitative analysis

Table 3 provides a sample of (query, passage) pairs from MSMARCO-Passage dev set, where there is a large discrepancy between the CA and DE model scores. Specifically, we consider pairs that the CA model scores low, but the DE model scores high. Interestingly, these pairs typically involve strong token overlap between the query and passage — indicating a certain degree of topicality — but are fact genuine negative pairs. This reflects that the DE model may be unable to capture certain fine-grained distinctions.

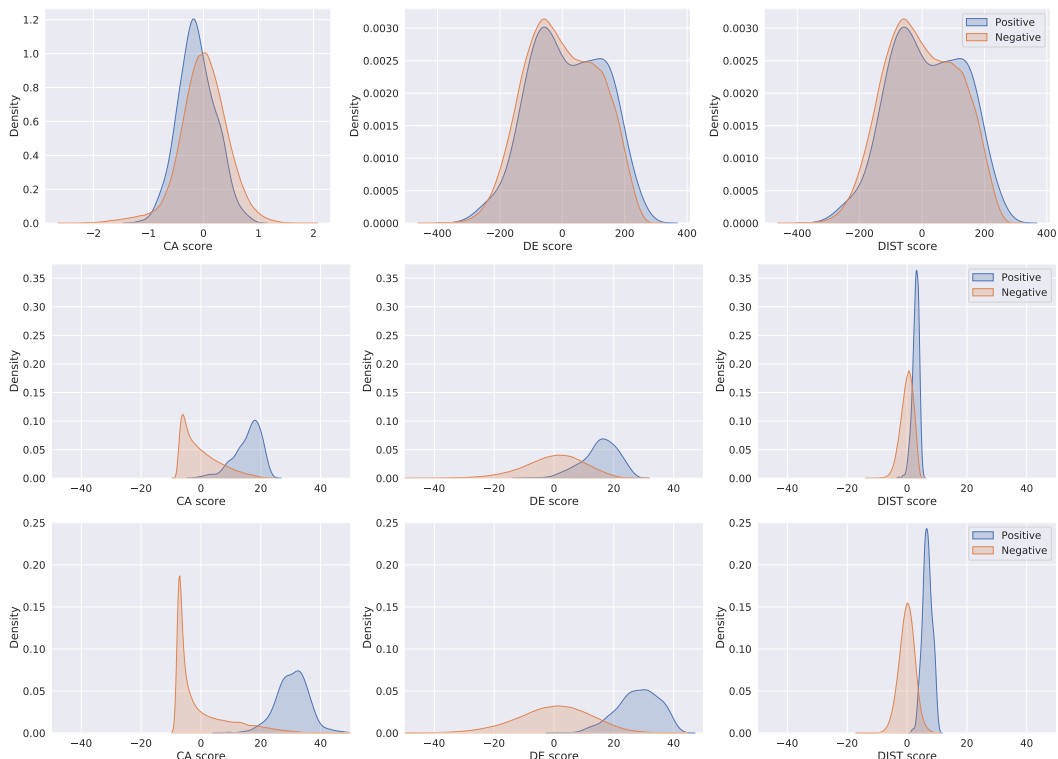

Figure 7: Comparison of CA, DE and distilled DE (DIST) model scores on the MSMARCO-Passage dev set at initialisation (top row), after $10,000$ steps of training (middle row), and at the completion of $300,000$ steps of training (bottom row). At initialisation, all models have overlapping score distributions, but the CA model operates at a much tighter range. After a few steps of training, the CA model already sees a peaky score distribution on the negative, while the DE model has a diffuse distribution. The distilled model however manages to overcome this, and produce a sharper distribution than even CA.

| Query | Passage |
|---|---|
| effects of yeast on body | The Side Effects of Chemotherapy on the Body. Chemotherapy drugs are powerful enough to kill rapidly growing cancer cells, but they also can harm perfectly healthy cells, causing side effects throughout the body. |
| age not to take off shoes at airline security | Airline Identification Requirements. Airlines do not typically require identification from passengers under the age of 18, but there are exceptions. Children under the age of 2 may ride on a parent's lap without purchasing a ticket, but the airline will require identification, such as a birth certificate, to prove the child's date of birth . . . |
| actress who plays alice on the magicians | Then portrayed as the animated Alice's real life counterpart by actress Mia Wasikowska as a more mature, grown up Alice in Disney's 2010 semi-sequel, live action/CGI film Alice and Wonderland Directed by Tim Burton. In the Broadway musical version, she will be played by Taylor Louderman . . . |
| can you absorb metals from plants | Answer 1: Photosynthesis is the ability of plants to absorb the energy of light, and convert it into energy for the plant. To do this, plants have pigment molecules which absorb the energy of light very well. The pigment responsible for most light-harvesting by plants is chlorophyll, a green pigment.The green color indicates that it is absorbing all the non-green light– the blues ( 425-450 nm), the reds and yellows (600-700 nm).he pigment responsible for most light-harvesting by plants is chlor . . . |

Table 3: Sample of (query, passage) pairs from MSMARCO-Passage dev set with largest discrepancy between the CA and DE model scores. In most of these cases, the passage is not relevant to the query; however, there is a high degree of token overlap between the two, indicating superficial similarity.

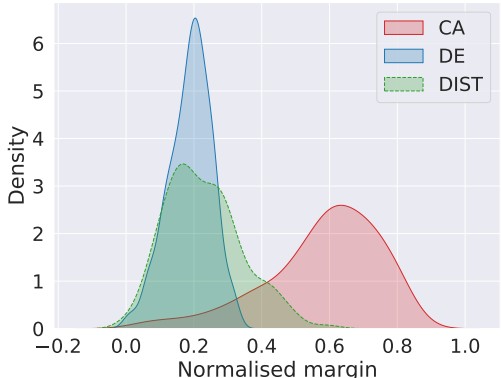

Figure 8: M³SE distillation model margins on the MSMARCO-Passage test set. Compared to the DE model trained with one-hot labels, the distilled model is seen to more confidently distinguish positives from negatives, evidenced by the margin distribution having more mass on larger values.

| Model | Supervision | Loss | MRR@10 | nDCG@10 |
|---|---|---|---|---|
| Cross-attention | One-hot | Softmax CE | 0.370 | 0.430 |
| Dual-encoder | One-hot | Softmax CE | 0.249 | 0.299 |
| Dual-encoder | Teacher | MSE | 0.180 | 0.232 |
| | Teacher | Margin MSE | 0.251 | 0.299 |
| | Teacher | KL | 0.273 | 0.321 |
| | Teacher | M³SE | 0.261 | 0.308 |

Table 4: Re-ranking results on MSMARCO-Passage dev set, using a small-bert-6-128 model.

## C.5 IMPACT OF DISTILLATION: EFFECT ON SCORE DISTRIBUTIONS

We have previously seen in Figure 7 that our proposed M³SE distillation strategy can improve the distribution of scores for positives and negatives. Figure 8 shows the impact that such distillation has on the margins between positive and negatives, compared to standard training on one-hot labels. The distilled model is seen to more confidently distinguish positives from negatives, as evidenced by the margin distribution having more mass on larger values.

## C.6 IMPACT OF DISTILLATION: EFFECT OF STUDENT MODEL ARCHITECTURE

The preceding distillation results have all employed a small-bert-6-768 model. It is of interest how sensitive the results are to this choice. To study this, we report in Table 4 the re-ranking results on MSMARCO-Passage when using a small-bert-6-128 model. We observe largely consistent trends as before: distillation using our proposed losses can close the gap between CA and DE model performance. Table 5 repeats this for a small-bert-2-768 model.

## C.7 IMPACT OF DISTILLATION: EFFECT OF NUMBER OF DOCUMENTS IN SUPERVISION

The results reported in the body employed a total of 20 documents per query for the softmax CE and M³SE methods. As this involves significantly more information than the triplet data used, e.g., in training the margin MSE loss, it is of interest to tease apart how much of the gains from the method come from this increased supervision, versus the loss itself. Table 6 studies the effect of varying the number of documents. We see that there are diminishing returns beyond a certain point: using 100 documents yields qualitatively similar performance to using 20 documents. At the same time, there is a marked difference in performance when using 2 documents versus 10.

| Model | Supervision | Loss | MRR@10 | nDCG@10 |
|-------|-------------|------|--------|---------|
| Cross-attention | One-hot | Softmax CE | 0.370 | 0.430 |
| Dual-encoder | One-hot | Softmax CE | 0.281 | 0.331 |
| Dual-encoder | Teacher | MSE | 0.259 | 0.313 |
| | Teacher | Margin MSE | 0.313 | 0.368 |
| | Teacher | KL | 0.311 | 0.370 |
| | Teacher | M$^3$SE | 0.286 | 0.337 |

Table 5: Re-ranking results on MSMARCO-Passage dev set, using a small-bert-2-768 model.

| # of documents | Loss | MRR@10 | nDCG@10 |
|----------------|------|--------|---------|
| 2 | Softmax CE | 0.327 | 0.386 |
| | M$^3$SE | 0.323 | 0.379 |
| 10 | Softmax CE | 0.333 | 0.391 |
| | M$^3$SE | 0.326 | 0.382 |
| 20 | Softmax CE | 0.338 | 0.396 |
| | M$^3$SE | 0.349 | 0.406 |
| 100 | Softmax CE | 0.334 | 0.393 |
| | M$^3$SE | 0.325 | 0.381 |

Table 6: Sensitivity analysis of distillation losses to number of documents per sample.

## C.8    RESULTS WITH COLBERT SCORER

Table 7 presents results using the ColBERT scorer. We find that with distillation using the softmax CE loss (Equation 1), we can *exceed* the performance of the cross-attention teacher model for the re-ranking task. This further indicates the viability of dual-encoder models for neural ranking.

| | MSMARCO rerank | | TREC DL19 | |
|---|---|---|---|---|
| | MRR@10 | nDCG@10 | MRR@10 | nDCG@10 |
| **Baselines: one-hot** | | | | |
| Cross-attention BERT | 0.370 | 0.430 | 0.829 | 0.749 |
| Dual-encoder ColBERT | 0.356 | 0.416 | 0.839 | 0.703 |
| **Distilled ColBERT: prior work** | | | | |
| MSE (Hofstätter et al., 2020a) | 0.365[†] | 0.428[†] | — | — |
| Margin-MSE (Hofstätter et al., 2020a) | 0.370[†] | 0.431[†] | 0.862[†] | 0.738[†] |
| **Distilled ColBERT: this work** | | | | |
| M$^3$SE | 0.371 | 0.430 | 0.875 | 0.726 |
| Softmax CE | 0.376 | 0.437 | 0.835 | 0.737 |

Table 7: Results on MSMARCO Passage dev set with ColBERT model.

## C.9    BEYOND DISTILLATION? EFFECT OF LABEL SMOOTHING

A special case of distillation is label smoothing (Szegedy et al., 2016), which can be understood as using a teacher that predicts the uniform distribution over all labels, and combining the result

| Smoothing $\alpha$ | MRR@10 train | MRR@10 test |
|---|---|---|
| 0.0 | 0.394 | 0.308 |
| 0.1 | 0.460 | 0.297 |
| 0.2 | 0.455 | 0.296 |
| 0.3 | 0.453 | 0.292 |
| 0.9 | 0.382 | 0.292 |
| 0.99 | 0.305 | 0.278 |

Table 8: Effect of label smoothing on DE models, on the re-ranking task for the MSMARCO-Passage dataset. For all models, we use the softmax CE loss on triplet data.

with the one-hot label. Specifically, this involves mixing the observed one-hot labels with a uniform distribution over all labels, with the mixing controlled by weight $\alpha \in [0, 1)$; $\alpha = 0$ corresponds to standard one-hot training. Label smoothing has also proven effective as a means of preventing harmful overfitting, and thus improving generalisation (Müller et al., 2019; Lukasik et al., 2020) in classification settings.

For the MSMARCO-Passage dataset, we studied the effect of varying degrees of label smoothing for DE models, using the softmax CE loss on triplet data. We use a small-bert-6-768 model as before; for computational ease, we report results after $300,000$ steps of training, at which point all models see stable test error. Table 8 reveals that, surprisingly, smoothing has a minimal effect on test performance, even at high smoothing levels. This is despite smoothing reducing the training performance at higher values, which is expected given its equivalence to injecting symmetric label noise to each sample. Further study of the viability of smoothing for retrieval problems would be of interest.

### C.10 BEYOND DISTILLATION? EFFECT OF ALTERNATE REGULARISATION STRATEGIES

Distillation provides one means of preventing the DE model from overfitting to the training set. One may of course conceive of other plausible strategies, such as:

- increasing the dropout rate in the final (embedding) layer of the transformer. The experiments in the body employ 10% dropout (following prior work, e.g., Hofstätter et al. (2020a)), which improves performance considerably over having no dropout; it is thus of interest whether there are benefits to further increasing this.

- performing token dropout at the input layer. Intuitively, such corruption prevents the model from overly relying on individual tokens, thus preventing spurious token matching (as present in some cases in Table 3).

- adding a masked language model (Devlin et al., 2019) loss for the student. This loss is a core component of BERT, and has thus proven to be effective in learning generically useful sequence embeddings. Intuitively, adding such a self-supervised objective to training can plausibly improve the model's robustness.

- modifying the base loss from square or log-loss to the focal loss (Lin et al., 2017). This loss has proven effective in vision tasks involving high class-imbalance. In our setting, as there are typically only a handful of relevant documents, but several millions of irrelevant documents, it is of interest whether such a loss can positively impact results.

Table 9 summarises results for each of these strategies on the MSMARCO Passage re-ranking task. Here, we focus on their effect on the standard (one-hot) training of a dot-product based DE model on triplet data. Unfortunately, we find that *none* of these strategies significantly improve the generalization performance of the baseline model. Thus, mitigating the overfitting observed in Figure 1 requires more effort than appealing to standard strategies effective in classification tasks.

### C.11 ANALYSIS OF TREC DL19 PREDICTIONS

Table 1 suggests a notable gap in performance for our proposed RankDistil variant and the margin MSE loss (Hofstätter et al., 2020a). Here, we take a closer look at the loss cases for our method. The

| Strategy | Train MRR@10 | Test MRR@10 |
|---|---|---|
| Baseline DE | 0.619 | 0.310 |
| Increased embedding dropout | 0.588 | 0.299 |
| Token dropout | 0.572 | 0.291 |
| Masked language loss | 0.548 | 0.299 |
| Focal loss | 0.546 | 0.307 |

Table 9: Results of various regularisation strategies on MSMARCO Passage dev set. For all rows, we use a DE model trained on the triplet data.

TREC DL19 data comprises a total of $43$ queries, each with between $\sim 100$ to $500$ rated passages. Of these queries, the MRR@10 of the DE models trained with margin MSE and softmax CE agree on $10$. On the queries with disagreement, $6$ cases favour margin MSE, and $4$ cases favour RankDistil-SM.

For the $6$ queries where RankDistil-SM ostensibly underperforms, we inspect the top-5 scoring passages for both margin MSE and softmax CE in Table 10 and 11. The cells shaded blue correspond to passages rated positive. Several other passages are however seen to be equally valid answers to the source query. Indeed, we submit that in *all* cases, the predictions from RankDistil are of at least the same quality as the margin MSE.

### C.12 IMPACT OF NEGATIVE MINING ON RE-RANKING PERFORMANCE

We consider the value of adding additional negatives during DE model training. Following Karpukhin et al. (2020); Qu et al. (2021), we consider the use of within-batch (also referred to as in-batch) and uniform negatives. For the latter, in each training minibatch, we draw a sample of $B_{\text{uni}}$ documents drawn uniformly at random from the entire set of documents; each such document is treated as a negative document in the loss. For the former, from each training minibatch $\{(q_i, \mathbf{d}_i, \mathbf{y}_i)\}_{i=1}^{B}$, we compute the set of all observed documents $\mathcal{D}_{\text{obs}} = \cup_{i=1}^{B} \cup_{k=1}^{K} \{d_{ik}\}$. For each sample $(q_i, \mathbf{d}_i, \mathbf{y}_i)$, we then use each element in $\mathcal{D}_{\text{obs}} - \cup_{k=1}^{K} \{d_{ik}\}$ as a negative document in the loss.

Table 12 summarises the results for the MSMARCO re-ranking task. We find that adding these negatives has a small gain for the one-hot model performance. However, for the distilled objectives, there is limited gain (and sometimes even a degradation) in performance. More study of this issue is warranted, but one hypothesis is that the re-ranking task is inherently concerned with ranking the outputs of a BM25 model. These outputs are precisely used to construct the training data used for distillation. It is thus possible that adding additional negatives — which are unlikely to appear as candidates for re-ranking — do not bring significantly useful information.

We emphasise here also that the results reported are for the *re-ranking*, as opposed to *retrieval* task. For the latter, adding within-batch and uniform negatives is intuitively and empirically valuable, as noted in Karpukhin et al. (2020); Qu et al. (2021). For example, in training the one-hot DE model on the triplet data, the retrieval MRR@10 is $0.281$. When adding within-batch and uniform negatives, this increases to $0.314$. Given that the re-ranking MRR@10 remains relatively unchanged ($0.312$ versus $0.310$), this is further indication of the re-ranking and retrieval objectives not being perfectly aligned.

| Query | Top scoring passages | |
| --- | --- | --- |
| | **Margin MSE** | **RankDistil-SM** |
| who is robert gray | Who is Henry Gray? Henry Gray is an African-American blues piano player and singer. He has been play... | Robert Grey (born 21 April 1951 in Marefield, Leicestershire) is an English musician best known as t... |
| | Robert Gray was the Democratic candidate for governor of Mississippi in the 2015 elections. Gray won... | Who is Henry Gray? Henry Gray is an African-American blues piano player and singer. He has been play... |
| | Kenneth Gray (I) Kenneth Gray is an actor, known for Love, Lies and Murder (1991), Retribution (1987... | Robert Gray was the Democratic candidate for governor of Mississippi in the 2015 elections. Gray won... |
| | Kenneth Gray (I) Actor. Kenneth Gray is an actor, known for Love, Lies and Murder (1991), Retributio... | Robert Gray. A surprise came on the Democratic side in the race for Mississippi Governor. Robert Gra... |
| | Robert Grey (born 21 April 1951 in Marefield, Leicestershire) is an English musician best known as t... | William Thomas Gray. Billy Gray was born on January 13, 1938 in Los Angeles, California, USA as Will... |
| define visceral? | Definition of visceral. 1 : felt in or as if in the internal organs of the body : deep a visceral co... | Definition of Visceral. Visceral: Referring to the viscera, the internal organs of the body, specifi... |
| | Definition of Visceral. Visceral: Referring to the viscera, the internal organs of the body, specifi... | Definition of Visceral. Visceral: Referring to the viscera, the internal organs of the body, specifi... |
| | Definition of visceral. 1 1 : felt in or as if in the internal organs of the body : deep a visceral... | Medical Definition of Visceral. Visceral: Referring to the viscera, the internal organs of the body,... |
| | Definition of Visceral. Visceral: Referring to the viscera, the internal organs of the body, specifi... | Definition of visceral. 1 : felt in or as if in the internal organs of the body : deep a visceral co... |
| | Definition of visceral. 1 1 : felt in or as if in the internal organs of the body : deep a visceral... | Define visceral: felt in or as if in the internal organs of the body : deep; not intellectual : inst... |
| what is the daily life of thai people | T he following concepts are part of Thai everyday life: or JAI YEN is more a way ... | The Daily Life of a Thai Monk The Sangha World in Thailand consists of about 200,000 monks and 85,00... |
| | The following concepts are part of Thai everyday life: or JAI YEN is more a way o... | An important thing in everyday life is SANUK. Thai people love to have fun together. SANUK can repre... |
| | The population of Thailand is approximately 67.5 million people, with an annual growth rate of about... | T he following concepts are part of Thai everyday life: or JAI YEN is more a way ... |
| | For the rapcore band, see Every Day Life. Everyday life or Daily life or Routine life is a phrase us... | The following concepts are part of Thai everyday life: or JAI YEN is more a way o... |
| | Everyday life. Everyday life, daily life or routine life comprises the ways in which people typicall... | The population of Thailand is approximately 67.5 million people, with an annual growth rate of about... |

Table 10: Comparison of top-5 scoring passages for margin MSE and softmax CE on TREC DL19 test set. The cells shaded blue correspond to passages rated positive. Several other passages are however seen to be equally valid answers to the source query.

| Query | Top scoring passages | |
| --- | --- | --- |
| | **Margin MSE** | **RankDistil-SM** |
| what are the three percenters? | The Three Percenters, formed in late 2008, are a loosely organized movement centered around an obscu... | Try to understand that being a Three Percenter (Threeper, 3%, 3 Percenter, etc) is more of an idea t... |
| | III% Club Merchandise & III% Logo Gear. The Three Percenters Club official merchandise is designed a... | But this still doesnt answer the question. So how do you know if you are a Three Percenter? Try t... |
| | Rhodes has written supportively of the Three Percenters, while at least two participants carried the... | So let me see if I can help. In fact, let me provide you with well over 50 ways to tell whether or n... |
| | Try to understand that being a Three Percenter (Threeper, 3%, 3 Percenter, etc) is more of an idea t... | The Three Percenters, formed in late 2008, are a loosely organized movement centered around an obscu... |
| | But this still doesnt answer the question. So how do you know if you are a Three Percenter? Try t... | III% Club Merchandise & III% Logo Gear. The Three Percenters Club official merchandise is designed a... |
| how are some sharks warm blooded | Most sharks are cold-blooded. Some, like the Mako and the Great white shark, are partially warmblood... | Most sharks are cold-blooded. Some, like the Mako and the Great white shark, are partially warmblood... |
| | Most sharks are cold-blooded. Some, like the Mako and the Great white shark, are partially warmblood... | Most sharks are cold-blooded. Some, like the Mako and the Great white shark, are partially warmblood... |
| | Are White Sharks warm-blooded or cold-blooded? White sharks are part of the fish family, so they mus... | Are White Sharks warm-blooded or cold-blooded? White sharks are part of the fish family, so they mus... |
| | Great white sharks are some of the only warm blooded sharks. This allows them to swim in colder wate... | Great white sharks are some of the only warm blooded sharks. This allows them to swim in colder wate... |
| | These sharks can raise their temperature about the temperature of the water; they need to have oc... | The Salmon Shark is one of the warmest of the warm-bodied sharks. Measurements of its epaxial (upper... |
| what are the social determinants of health | The social determinants of health are the circumstances in which people are born, grow up, live, wor... | Social determinants of health reflect the social factors and physical conditions of the environment ... |
| | Social determinants of health reflect the social factors and physical conditions of the environment ... | Social determinants of health are economic and social conditions that influence the health of people... |
| | The social determinants of health are linked to the economic and social conditions and their distrib... | Social determinants of health are conditions in the environments in which people are born, live, lea... |
| | Social determinants of health are conditions in the environments in which people are born, live, lea... | Back to Top. Social determinants of health are conditions in the environments in which people are bo... |
| | Determinants of health are factors that contribute to a person's current state of health. These fact... | The social determinants of health are the circumstances in which people are born, grow up, live, wor... |

Table 11: Comparison of top-5 scoring passages for margin MSE and softmax CE on TREC DL19 test set. The cells shaded blue correspond to passages rated positive. Several other passages are however seen to be equally valid answers to the source query.

| Supervision | Loss | MRR@10 |
|---|---|---|
| One-hot | Softmax CE | 0.310 |
| | + negative mining | 0.312 |
| Teacher | Margin MSE | 0.334 |
| | + negative mining | 0.335 |
| Teacher | M$^3$SE | 0.349 |
| | + negative mining | 0.324 |
| Teacher | KL | 0.346 |
| | + negative mining | 0.342 |

Table 12: Results of negative mining for DE models, on MSMARCO passage re-ranking task.

