# OpenReview forum: "In defense of dual-encoders for neural ranking"
_ICLR.cc/2022/Conference — ICLR 2022 Submitted_

### Official Review · Reviewer_4c6M · 2021-10-29

**Correctness:** 4
**Technical Novelty And Significance:** 2
**Empirical Novelty And Significance:** 2
**Recommendation:** 3
**Confidence:** 5

**Main Review:**

Strengths:
The paper is well written and easy to follow.  Good analysis throughout, comparing the score distributions of the two architectures.  The experiments are sound and proofs are correct.

Weak points:
The theory presented is correct, but not really informative in practice.  Prop. 1 has no bounds on the dimension of the embeddings.  Without such bound, it's easy to see how bi-encoders might represent any function, however this is not really useful.  I don't think this theory deserves the central treatment it currently has in the presentation, I would remove or relegate to the appendix.  On the other hand, it's interesting to see empirically that bi-encoders have similar training accuracy with cross-encoders.

Novelty - there are multiple recent works which have demonstrated the usefulness of distilling a cross-encoder into a bi-encoder for retrieval.  Authors have cited some of them, but here are two more:
https://arxiv.org/pdf/2010.10999.pdf
https://arxiv.org/pdf/2010.08191.pdf

All of these works use the standard cross-entropy / KL-divergence loss for distillation, which the authors have shown to work as well or better than their proposed M3SE loss.  From a practical standpoint, the novel contribution of this work is pretty weak.

As mentioned before, some of the empirical analysis is certainly interesting, but I don't think it is surprising enough to warrant publication by itself.

**Summary Of The Paper:**

The paper contrasts cross-encoder vs. bi-encoder architectures for re-ranking in information retrieval.  The authors argue that bi-encoders under-perform cross-encoders, not for lack of capacity, but due to poor generalization, and back this claim with some theory and empirical results.  The authors then propose distillation of the cross-encoder model into a bi-encoder, which is shown to improve bi-encoder results.  While this has been proposed before, the distillation loss function the authors propose is slightly different than the standard cross-entropy (which the authors show is closely related).

**Summary Of The Review:**

This is a well written paper, however the contribution is not significant, due to multiple recent works already having covered this idea pretty well.  The theory presented is also neither informative nor surprising.  Some of the empirical analysis is certainly interesting, but I don't think it is surprising enough to warrant publication by itself.

---

> ### Author Response · Authors · 2021-11-17
> **Response to Reviewer 4c6M**
>
> Thanks for the detailed comments.
>
> > From a practical standpoint, the novel contribution of this work is pretty weak.
>
> We wish to emphasise a few points:
>
> - The primary aim of this work is in illuminating the nature of the gap between the CA and DE performance (cf. para before Section 2). While the _existence_ of this gap has been widely noted in prior work, the _reason_ it exists has not (to our knowledge) been systematically studied.
>
>
> - Our main finding is that that the gap is mostly an issue of generalisation rather than capacity (Figure 1), which in turn can be traced to poorer score distributions for DE models (Figure 2). We believe these are conceptually and practically useful observations, as they motivate spending effort in addressing this generalisation gap, in particular by ensuring a better separation of scores.
>
>
> > The theory presented is correct, but not really informative in practice. Prop. 1 has no bounds on the dimension of the embeddings. Without such bound, it's easy to see how bi-encoders might represent any function, however this is not really useful. I don't think this theory deserves the central treatment it currently has in the presentation, I would remove or relegate to the appendix.
>
> Proposition 1 does not intend to suggest a practical algorithm. Rather, it simply aims to explain the phenomenon in Figure 1, wherein the _training_ performance of the DE model (with suitably large embedding size) is on-par with the CA model. The Proposition shows that this finding can be explained by the high theoretical capacity of these models.
>
> We remark also that this analysis occupies only one out of several sections devoted to studying the gap between CA and DE models.
>
> > Novelty - there are multiple recent works which have demonstrated the usefulness of distilling a cross-encoder into a bi-encoder for retrieval. Authors have cited some of them, but here are two more: https://arxiv.org/pdf/2010.10999.pdf https://arxiv.org/pdf/2010.08191.pdf
>
> Thanks for the references. Please note that *we do cite* (Qu et al., ‘21) (above Equation 2). We have added the citation to (Yang and Seo, ‘20), which we think is well aligned with the works cited in Section 2.
>
>
> > All of these works use the standard cross-entropy / KL-divergence loss for distillation, which the authors have shown to work as well or better than their proposed M3SE loss.
>
> Please note that KL divergence does *not* work as well or better than M3SE in our experiments. Indeed, on both the MSMARCO and NQ datasets, M3SE is consistently (if sometimes only slightly) superior. The _only_ case where we observed KL to be clearly superior is on the nDCG metric for the TREC DL19 set. Further, as noted in Appendix C.11, many of these cases appear to be the result of rating noise.
>
>
> In Section 2, we did state that distilling with the KL divergence has been previously studied, with citations to relevant prior work. To reduce confusion, we have re-iterated this point in Section 4.2. However, the connection to the margin MSE loss (Proposition 3) has not been noted, nor has the loss been justified as preserving the teacher’s ranking over documents (Section 4.2).

---

### Official Review · Reviewer_oYj4 · 2021-10-31

**Correctness:** 4
**Technical Novelty And Significance:** 3
**Empirical Novelty And Significance:** 3
**Recommendation:** 6
**Confidence:** 2

**Main Review:**

Strengths:
* They did a theoretical analysis which shows that DE models are sufficiently expressive to model a broad class of scores and there should not be a gap between CA and DE models.
* They identified that the gap is mainly caused by the generalization ability of the DE models. They have similar performance as CA models on the training data while performing much worse on the testing data.
* They proposed distillation algorithms that aim to match the margins between the teacher (CA) and student (DE) models. The experimental results verify that the proposed approach can further reduce the gap.

Weaknesses:
* It remains unclear to me why the DE models are less generalizable. Do the authors have some any explanations about this?
* The theoretical analysis can not guide the algorithm design to reduce the gap.
* They did not compare with other distillation algorithms.

**Summary Of The Paper:**

This paper investigate the gap problem between cross-attention (CA) and dual-encoder (DE) models for document reranking. In general, cross-attention models perform much better than dual-encoder models while consuming more computational cost. The authors first leverage Mercer's theorem to prove that dual-encoders models are sufficiently expressive to model a broad class of scores using countably infinite dimension, which mainly the gap between CA and DE models should not exist from theoretical analysis. They further identified the gap is caused by the generalization ability of the DE models. To reduce the overfitting on the training data, they proposed to use distillation algorithms. Specially, they design the loss function which matches the margins between the teacher (CA) and student (DE) models. They also extend the margin to probability matching by using softmax cross-entropy loss. Through the experiments, they demonstrated that their proposed distillation approaches can further reduce the gap between DE and CA models.

The main contributions of this paper is that they show there should not be a gap between DE and CA models from theoretical analysis. They found that the gap is caused by the generalization ability of CA model and proposed distillation approach to reduce the gap.

**Summary Of The Review:**

They have theoretically analyzed that the DE models are expressive enough and the gap should not exist between DE and CA models. They found the gap is because DE models are less generalizable and proposed an effective distillations algorithm to further reduce the gap. Taking these advantages into consideration, I think it is good paper to be accepted.

---

> ### Author Response · Authors · 2021-11-17
> **Response to Reviewer oYj4**
>
> Thanks for the detailed comments. We are glad the reviewer finds this a “good paper to be accepted”.
>
> > It remains unclear to me why the DE models are less generalizable. Do the authors have some any explanations about this?
>
> Section 3.3 includes some relevant discussion on this point. Intuitively, the DE model strongly ties together the parameters for all (query, document) pairs sharing a query or document. As a result, the model tends to produce more diffuse score distributions, which do not confidently separate the positive from negative documents (Figure 2, left and middle panel). By distilling from the CA model, we can however improve the separation (Figure 2, right panel).
>
>
> > The theoretical analysis can not guide the algorithm design to reduce the gap.
>
> The theoretical analysis aims to establish that DE models have sufficient capacity to model scores. The empirical results in Figure 1 point to their poorer performance as being a result of how these models are _trained_. Further, the results in Figure 2 indicate that a key reason for the DE model under-performance is the lack of sufficient _margins_. Our proposed distillation techniques seek to achieve exactly this.
>
>
> > They did not compare with other distillation algorithms.
>
> We have compared against logit matching, margin MSE, and RankDistil distillation losses. We believe these are representative state-of-the-art distillation algorithms for this task. We can certainly include any other distillation baselines the reviewer has in mind.

---

> > ### Comment · Reviewer_oYj4 · 2021-11-29
> > **Thanks for the reply**
> >
> > Thanks for the reply. That answers my questions.

---

### Official Review · Reviewer_FgFj · 2021-11-02

**Correctness:** 3
**Technical Novelty And Significance:** 3
**Empirical Novelty And Significance:** 3
**Recommendation:** 5
**Confidence:** 3

**Main Review:**

Closing the gap between cross-attention models and dual-encoder models is an essential research problem for industrial applications. The main contribution of this work lies in revealing the underlying reasons of gap between CA and DE models, and proposing a new KD method for re-ranking tasks. The whole paper is clear and well-written,

Below are some questions or confusions I had:
1)	I am a bit confused by the relationships between capacity, overfitting and generalization ability (or better OOD performance) as discussed in this work. It is obvious that a model with relatively large capacity tends to easily overfit the training data, while to get better OOD performances we need to establish new learning methods or design better model architectures. The BERT model has a large number of parameters, so it is not surprising that a dual-encoder model, twice size of a CA model, can overfit the training dataset, which is widely accepted in practice. To narrow the gap, the proposed is a modified KD loss, and it would be more interesting to give some suggestions on the network architectures, as the well-known ColBERT.

2)    The main contribution of this work is the Dual-encoders(DE) model based distillation, and this topic has been studied a lot. The proposed M3SE loss substantially outperforms previous distillation techniques on both re-ranking and full retrieval settings. However, M3SE seems to be a natural extension of previous work MarginMSE by adapting to multiple positive and negative documents. Considering that there is almost one positive doc for each query in MS MARCO and NQ, Equation 4 actually reduces to $(s_{+}-s_{j}^{*})^2+\sum_{j \in N} \[s_j-s_{j}^{\*} \]^2_{+}$  in the experiment. This means that in M3SE, the teacher provides an benchmark for student model that what should the score of positive document be close to and what should its negative ones be lower to. It makes sense but seems to be a natural extension of marginal MSE.

3)	In this work, 6-layer BERTs are used as student model. The authors are suggested to further study the question of how to find the optimal student network architectures, e.g. a relatively deep and thin network, under the constraints of inference time.


**Summary Of The Paper:**

This paper aims to narrow the performance gap between cross-attention BERT and dual-encoder BERT for re-ranking task. The authors empirically and theoretically analyze the underlying reasons of the performance gap. The gap could be mitigated by the proposed knowledge distillation method, where the original cross-attention BERT model acts as the “teacher” and the more efficient dual-encoder BERT model is used as student. Comprehensive experiments confirmed the effectiveness of the proposed KD method for various re-ranking tasks.

**Summary Of The Review:**

This paper tries to reveal the underlying reasons of performance gap between CA and DE models, and a KD method is further proposed to improve the performance of dual-encoder model, while the analysis on the gap cannot guide the design of new student matching architecture or new distillation method.

---

> ### Author Response · Authors · 2021-11-17
> **Response to Reviewer FgFj**
>
> Thanks for the detailed comments.
>
> > I am a bit confused by the relationships between capacity, overfitting and generalization ability (or better OOD performance) as discussed in this work. It is obvious that a model with relatively large capacity tends to easily overfit the training data, while to get better OOD performances we need to establish new learning methods or design better model architectures.
>
> We would like to emphasise a few points:
>
> - We use the term “high capacity” to refer to the existence of a suitable model configuration that can mimic an arbitrarily complex score relationship between queries and documents.
>
>
> - While DE models can have a large number of parameters, this alone does not necessarily imply they have high capacity. One needs to establish that the way these parameters are _used_ – i.e., the factorised score in Equation 2 – is sufficiently powerful. To illustrate this point, consider a variant of the DE model class, which produces scores via $w_1^\top {\rm pool}(T(q)) + w_2^\top {\rm pool}(T(d))$ for learned weights $w_1, w_2$. Such a model can have a large number of parameters, but will be unable to model scores that involve any interaction between the queries and documents.
>
>
> - CA models have similarly high capacity as DE models. However, they manage to perform well on train _and_ test samples. (See also the below point.) By contrast, DE models only perform well on train samples. This implies that there are certain issues with the training of DE models, which motivates ways of improving the same (e.g., distillation).
>
>
> > The BERT model has a large number of parameters, so it is not surprising that a dual-encoder model, twice size of a CA model, can overfit the training dataset, which is widely accepted in practice.
>
> Please note that our dual-encoders use tied weights for the query and documents (see page 8). Thus, the number of parameters for the two models is comparable.
>
> On this point, we also observed that using un-tied weights generally results in slightly worse performance, as has been previously noted in [Hofstatter et al., ‘20]. This further illustrates that having more parameters is not necessarily better for final generalisation performance.
>
> > To narrow the gap, the proposed is a modified KD loss, and it would be more interesting to give some suggestions on the network architectures, as the well-known ColBERT.
>
> Architecture modification is certainly an interesting route, which we did discuss as a valid possibility in Section 3.4. Our focus in this paper was primarily on exploring the limits of dot-product based scoring. The motivation for trying to maximise the performance of a dot-product scorer is that the latter involves significantly lower memory and/inference burden than extensions such as ColBERT (see, e.g., Table 1 of Hofstatter et al. ‘20).
>
> > However, M3SE seems to be a natural extension of previous work MarginMSE by adapting to multiple positive and negative documents. Considering that there is almost one positive doc for each query in MS MARCO and NQ, Equation 4 actually reduces to (s+−sj∗)2+∑j∈N[sj−sj∗]+2 in the experiment. It makes sense but seems to be a natural extension of marginal MSE.
>
> We certainly agree that this idea is intuitive, but view this as a virtue of the method. To our knowledge, this idea has not been systematically explored. Further, as noted in Sec 4.3, this objective connects to the recently proposed RankDistil objective of [Reddi et al., ‘21].
>
> > In this work, 6-layer BERTs are used as student model. The authors are suggested to further study the question of how to find the optimal student network architectures, e.g. a relatively deep and thin network, under the constraints of inference time.
>
>
> The use of a 6-layer BERT student follows previous work, e.g., [Hofstatter et al., ‘20]. In Appendix C.1, we did consider the effect of varying the student architecture in terms of embedding size. We have added results when distilling to a 2-layer BERT model in Appendix C.7.
>
> We completely agree that studying the design of an optimal student under a fixed inference budget is of practical interest. However, we believe that such study is complementary to the primary concern of this work – viz., understanding the reasons for the gap between CA and DE performance – and would thus be suitable for a separate study. Indeed, we do note that recent works (e.g., Rawat et al., ‘21, “When in Doubt, Summon the Titans: Efficient Inference with Large Models”) have studied how distillation can aid in this task.

---

> > ### Comment · Reviewer_FgFj · 2021-11-29
> > **Thanks for your reply**
> >
> > Thanks for your reply and efforts! Your reply clarified my confusions on the capacity of DE models.

---

### Official Review · Reviewer_yXc2 · 2021-11-04

**Correctness:** 3
**Technical Novelty And Significance:** 3
**Empirical Novelty And Significance:** 3
**Recommendation:** 6
**Confidence:** 4

**Main Review:**



Pros:

1. Good theoretical analysis on the power of dual encoders. I like this analysis as the starting point of the paper.
2. The generalization gap of dual encoders makes sense to me. The empirical results in Figure 2 are interesting. However, I would like to see more deep insight. For example, analysis on special query-document cases might be useful.
3. The two modification methods on knowledge distillation are interesting. The empirical results show the benefits of the proposed methods. However, I have some questions about this part (refer to the Cons part).
.


Cons:

My main concern of the proposed approach is the experimental design parts. For information retrieval models, only utilizing the human-labeled dataset for knowledge distillation seems to be a weak baseline.  There has been some work on constructing datasets to improve the dual encoders (also called data distillation). For example, we can easily construct hard negative samples (cited in the paper) to improve the dual encoders. My feeling is that adding these results might make the paper even stronger.


**Summary Of The Paper:**

In information retrieval problems, there are two kinds of models: dual-encoder models and cross-attention models. Cross-attention models generally outperform dual encoder models by a large margin. Due to the latency requirement of real search systems, cross-attention models might not meet the requirements. So one ideal solution is to transfer the knowledge from cross-attention models to dual encoder models. This paper follows this line of research ideas by reducing the gap between the two models.

Firstly, the paper theoretically analyzes the effects with sufficiently large embedding dimensions. Secondly, the paper empirically verifies the theoretical results. Thirdly, the paper proposes a new method that further bridges the gap between the two methods. The writing is clear and well-structured. I like the idea and logic of the paper. I have some questions about the experiment part.


**Summary Of The Review:**

Overall, I like this paper. I would like more empirical results on data distillations.

---

> ### Author Response · Authors · 2021-11-17
> **Response to Reviewer yXc2**
>
> Thanks for the detailed comments. We are glad the reviewer appreciated the work.
>
> > I would like to see more deep insight. For example, analysis on special query-document cases might be useful.
>
> We do provide some qualitative analyses on (query, document) pairs in Table 3 (Appendix C.4), which shows cases where DE models produce very different scores compared to CA models. As discussed in Appendix C.4, several of these involve cases with high token overlap between the query and passage, i.e., the pairs are _superficially_ related. This is intuitive: the DE model strongly ties together the parameters for all (query, document) pairs sharing either a query or document. As a result, it is harder for the model to make more precise distinctions.
>
> > we can easily construct hard negative samples (cited in the paper) to improve the dual encoders
>
> As noted in the Conclusion, we do see systematic study of the use of hard negatives, and performance under the retrieval setting as an interesting avenue for future work. In the present work, we sought to compare the CA and DE models on an equal footing, which required the use of the re-ranking setting (as CA models are not applicable for retrieval).
>
> Nonetheless, as an illustrative example, we have added new experiments in Appendix C.12. These add within-batch (also known as in-batch) and uniform negatives during training. These were shown to be useful for _full retrieval_ performance in [Karpukhin et al. ‘20], [Qu et al. ‘21] (amongst others). We find that for _re-ranking_, these negatives appear to be neutral in terms of the MRR. One possible reason for this finding is that these tasks are somewhat different in nature: in re-ranking, the goal is to rank the outputs of a BM25 model. In the “triplet” training data which forms the basis for all methods considered, the documents considered are precisely generated based on the output of a BM25 model. Thus, the negatives in this data are already well-aligned with the final goal, and adding within-batch and uniform negatives may not bring further value.

---

### Decision · Program_Chairs · 2022-01-20

**Decision:**

Reject

**Comment:**

Strengths:
* Well-written paper
*Theoretical analysis demonstrates that dual encoder models have similar capacity as CA models
*New distillation algorithm for learning DE students from CA teachers

Weaknesses:
* No reviewer seems particularly excited about this work
* Theoretical analysis doesn’t provide actionable insight -- it does not directly motivate the suggested distillation methods
* Empirical results are lacking -- reviewers asked for qualitative examples of improvements from their distillation method